# An Online Learning Approach to Generative Adversarial Networks

**Paulina Grnarova, Kfir Y. Levy, Aurelien Lucchi, Thomas Hofmann, Andreas Krause**
ETH Zürich
{paulina.grnarova,yehuda.levy,aurelien.lucchi,thomas.hofmann}@inf.ethz.ch
krausea@ethz.ch

## Abstract

We consider the problem of training generative models with a Generative Adversarial Network (GAN). Although GANs can accurately model complex distributions, they are known to be difficult to train due to instabilities caused by a difficult minimax optimization problem. In this paper, we view the problem of training GANs as finding a *mixed* strategy in a zero-sum game. Building on ideas from online learning we propose a novel training method named Chekhov GAN [1]. On the theory side, we show that our method provably converges to an equilibrium for semi-shallow GAN architectures, i.e. architectures where the discriminator is a one-layer network and the generator is arbitrary. On the practical side, we develop an efficient heuristic guided by our theoretical results, which we apply to commonly used deep GAN architectures. On several real-world tasks our approach exhibits improved stability and performance compared to standard GAN training.

## 1 Introduction

A recent trend in generative models is to use a deep neural network as a generator. Two notable approaches are variational auto-encoders (VAE) (Kingma & Welling, 2013; Rezende et al., 2014) as well as Generative Adversarial Networks (GAN) (Goodfellow et al., 2014). Unlike VAEs, the GAN approach offers a way to circumvent log-likelihood-based estimation and it also typically produces visually sharper samples (Goodfellow et al., 2014). The goal of the generator network is to generate samples that are indistinguishable from real samples, where indistinguishability is measured by an additional discriminative model. This creates an adversarial game setting where one pits a generator against a discriminator.

Let us denote the data distribution by $p_{\text{data}}(\mathbf{x})$ and the model distribution by $p_{\mathbf{u}}(\mathbf{x})$. A probabilistic discriminator is denoted by $h_{\mathbf{v}} : \mathbf{x} \to [0;1]$ and a generator by $G_{\mathbf{u}} : \mathbf{z} \to \mathbf{x}$. The GAN objective is:

$$\min_{\mathbf{u}} \max_{\mathbf{v}} M(\mathbf{u}, \mathbf{v}) = \frac{1}{2}\mathbb{E}_{\mathbf{x} \sim p_{\text{data}}} \log h_{\mathbf{v}}(\mathbf{x}) + \frac{1}{2}\mathbb{E}_{\mathbf{z} \sim p_{\mathbf{z}}} \log(1 - h_{\mathbf{v}}(G_{\mathbf{u}}(\mathbf{z}))) \,. \tag{1}$$

Each of the two players (generator/discriminator) tries to optimize their own objective, which is exactly balanced by the loss of the other player, thus yielding a two-player zero-sum minimax game. Standard GAN approaches aim at finding a pure Nash Equilibrium by using traditional gradient-based techniques to minimize each player's cost in an alternating fashion. However, an update made by one player can repeatedly undo the progress made by the other one, without ever converging.

In general, alternating gradient descent fails to converge even for very simple games as shown by Salimans et al. (2016). In the setting of GANs, one of the central open issues is this non-convergence problem, which in practice leads to oscillations between different kinds of generated samples (Metz et al., 2016).

While standard GAN methods seek to find pure minimax strategies, we propose to consider mixed strategies, which allows us to leverage online learning algorithms for mixed strategies in large games.

---

[1]We base this name on the Chekhov's gun (dramatic) principle that states that every element in a story must be necessary, and irrelevant elements should be removed. Analogously, our Chekhov GAN algorithm introduces a sequence of elements which are eventually composed to yield a generator.

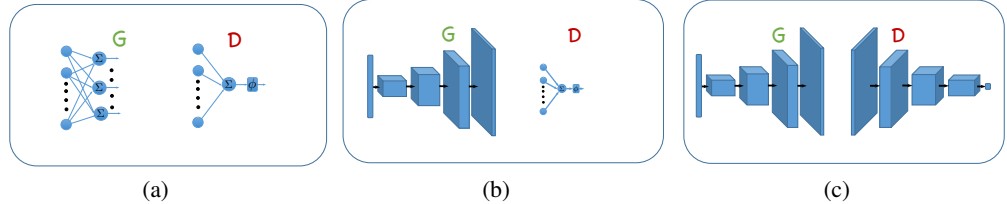

Figure 1: Three types of GAN architectures. Left: *shallow*. Middle: *semi-shallow*. Right: *deep*.

Building on the approach of Freund & Schapire (1999), we propose a novel training algorithm for GANs that we call CHEKHOV GAN .

The standard GAN training method is not guaranteed to converge for general GAN architectures. Nevertheless, it does converge for *shallow* ones [2], i.e. for GAN architectures which consist of a single layer network as a discriminator, and a generator with one hidden layer (see Fig. 1(a)). Unfortunately, *shallow* GANs are very different from the deep GANs (Fig. 1(c)) which are used in practice, and ideally one would hope to understand models which are more similar to deep architectures.

In this paper we make a step forward by considering *semi-shallow* GANs, an intermediate architecture where the generator is any arbitrary network and the discriminator consists of a single layer (Fig. 1(b)). Our contributions are:

**(1)** We show that finding a Mixed Nash Equilibrium (MNE or simply equilibrium) gives rise to useful generators and discriminators.

**(2)** We provide an algorithm that provably converges to an equilibrium for semi-shallow architectures.

**(3)** Guided by our theoretical results we devise a new GAN training algorithm that is applicable to standard *deep* GAN architectures.

We first discuss the benefits of pursuing a mixed equilibrium. Based on results from game theory we show that by reaching an equilibrium we obtain useful (mixed) generator and discriminator. In the context of GANs, "usefulness" means that the mixed generator provided by the equilibrium solution will "fool" any adversary at least as good as any single generator (similar results hold for the discriminator).

On the theory side, we show that GANs with *semi-shallow* architectures induce *semi-concave* games, i.e., games which are concave with respect to the max player, but need not have a special structure with respect to the min player. Then we show that in such games players may efficiently invoke regret minimization procedures in order to find an equilibrium; this in turn gives rise to a way of finding an equilibrium in *semi-shallow* GANs. To the best of our knowledge, this result is novel in the context of GANs and might also find uses in other scenarios where such structure may arise. We would like to emphasize that this is a significant step from a theoretical point of view as the standard approach to training GANs is only known to be theoretically sound for *convex-concave* games, which correspond to shallow-networks (though it is still used heuristically to train deep GAN architectures).

On the practical side, we develop an efficient heuristic guided by our theoretical results, which we apply to commonly used *deep* GAN architectures. We provide experimental results demonstrating that our approach exhibits better empirical stability compared to the vanilla GAN and generates more diverse samples, while retaining the same level of visual quality.

In Section 2, we briefly review necessary notions from online learning and zero-sum games. We then present our approach and its theoretical guarantees in Section 3, and our practical algorithm is presented in Section 4. Lastly, we present empirical results on standard benchmarks in Section 5.

---

[2]Note that the standard GAN training procedure, as is, does not converge to a Nash equilibrium for *shallow* architectures, see e.g. Salimans et al. (2016). However, averaging the models does yield convergence in this case.

## 2 Background & Related Work

### 2.1 GANs

**GAN Objectives:** The classical way to learn a generative model consists of minimizing a divergence function between a parametrized model distribution $p_{\mathbf{u}}(\mathbf{x})$ and the true data distribution $p_{\text{data}}(\mathbf{x})$. The original GAN approach by Goodfellow et al. (2014) is for example known to be related to optimizing the Jensen-Shannon divergence. This was later generalized by Nowozin et al. (2016) who described a broader family of GAN objectives stemming from $f$-divergences. A different popular type of GAN objectives is the family of Integral Probability Metrics (Müller, 1997), such as the kernel MMD (Gretton et al., 2012; Li et al., 2015) or the Wasserstein metric (Arjovsky & Bottou, 2017). All of these divergence measures yield a minimax objective.

**Training methods for GANs:** In order to solve the minimax objective in Eq. 1, Goodfellow et al. (2014) suggested an approach that alternatively minimizes over $\mathbf{u}$ and $\mathbf{v}$ using mini-batch stochastic gradient descent. This approach can be shown to converge only when the updates are made in function space. In practice, this condition is not met - since this procedure works in the parameter space - and many issues arise during training (Arjovsky & Bottou, 2017; Radford et al., 2015), thus requiring careful initialization and proper regularization as well as other tricks (Metz et al., 2016; Pfau & Vinyals, 2016; Radford et al., 2015; Salimans et al., 2016). Even so, several problems are still commonly observed including a phenomena where the generator oscillates, without ever converging to a fixed point, or mode collapse when the generator maps many latent codes $z$ to the same point, thus failing to produce diverse samples.

The closest work related to our approach is that of Arora et al. (2017) who showed the existence of an approximate mixed equilibrium with certain generalization properties; yet without providing a constructive way to find such equilibria. Instead, they advocate the use of mixed strategies, and suggest to do so by using the exponentiated gradient algorithm of Kivinen & Warmuth (1997). The work of Tolstikhin et al. (2017) also uses a similar mixture approach based on boosting. Other works have studied the problem of equilibrium and stabilization of GANs, often relying on the use of an auto-encoder as discriminator (Berthelot et al., 2017) or jointly with the GAN models (Che et al., 2016). In this work, we focus on providing convergence guarantees to a *mixed equilibrium* (definition in Section 3.2) using a technique from online optimization that relies on the players' past actions.

### 2.2 Online Learning

Online learning is a sequential decision making framework in which a player aims at minimizing a cumulative loss function revealed to her sequentially. The source of the loss functions may be arbitrary or even adversarial, and the player seeks to provide worst case guarantees on her performance. Formally, this framework can be described as a repeated game of $T$ rounds between a player $\mathcal{P}_1$ and an adversary $\mathcal{P}_2$. At each round $t \in [T]$:

1. The player ($\mathcal{P}_1$) chooses a point $\mathbf{u}_t \in \mathcal{K}$ according to some algorithm $\mathcal{A}$
2. The adversary ($\mathcal{P}_2$) chooses a loss function $f_t \in \mathcal{F}$
3. The player ($\mathcal{P}_1$) suffers a loss $f_t(\mathbf{u}_t)$, and the loss function $f_t(\cdot)$ is revealed to her.

The adversary is usually limited to choosing losses from a structured class of objectives $\mathcal{F}$, most commonly linear/convex losses. Also, the decision set $\mathcal{K}$ is often assumed to be convex. The performance of the player's strategy is measured by the *regret*, defined as,

$$\text{Regret}_T^{\mathcal{A}}(f_1, \ldots, f_T) = \sum_{t=1}^{T} f_t(\mathbf{u}_t) - \min_{\mathbf{u}^* \in \mathcal{K}} \sum_{t=1}^{T} f_t(\mathbf{u}^*) . \tag{2}$$

Thus, the regret measures the cumulative loss of the player compared to the loss of the *best fixed decision in hindsight*. A player aims at minimizing her regret, and we are interested in *no-regret* strategies for which players ensure regret which is *sublinear* in $T$ for any loss sequence [3].

---

[3]A regret which depends linearly on $T$ is ensured by any strategy and is therefore trivial.

While there are several no-regret strategies, many of them may be seen as instantiations of the *Follow-the-Regularized-Leader* (FTRL) algorithm where

$$\mathbf{u}_t = \arg\min_{\mathbf{u} \in \mathcal{K}} \sum_{\tau=1}^{t-1} f_\tau(\mathbf{u}) + \eta_t^{-1} R(\mathbf{u}) \qquad \textbf{(FTRL)} \tag{3}$$

FTRL takes the accumulated loss observed up to time $t$ and then chooses the point in $\mathcal{K}$ that minimizes the accumulated loss plus a regularization term $\eta_t^{-1} R(\mathbf{u})$. The regularization term prevents the player from abruptly changing her decisions between consecutive rounds[4]. This property is often crucial to obtaining no-regret guarantees. Note that FTRL is not always guaranteed to yield no-regret, and is mainly known to provide such guarantees in the setting where losses are linear/convex (Hazan et al., 2016; Shalev-Shwartz et al., 2012).

## 2.3 ZERO-SUM GAMES

Consider two players, $\mathcal{P}_1, \mathcal{P}_2$, which may choose pure decisions among the sets $\mathcal{K}_1$ and $\mathcal{K}_2$, respectively. A zero-sum game is defined by a function $M : \mathcal{K}_1 \times \mathcal{K}_2 \mapsto \mathbb{R}$ which sets the utilities of the players. Concretely, upon choosing a pure strategy $(\mathbf{u}, \mathbf{v}) \in \mathcal{K}_1 \times \mathcal{K}_2$ the utility of $\mathcal{P}_1$ is $-M(\mathbf{u}, \mathbf{v})$, while the utility of $\mathcal{P}_2$ is $M(\mathbf{u}, \mathbf{v})$. The goal of either $\mathcal{P}_1/\mathcal{P}_2$ is to maximize their worst case utilities; thus,

$$\min_{\mathbf{u} \in \mathcal{K}_1} \max_{\mathbf{v} \in \mathcal{K}_2} M(\mathbf{u}, \mathbf{v}) \quad \textbf{(Goal of } \mathcal{P}_1\textbf{)}, \quad \& \quad \max_{\mathbf{v} \in \mathcal{K}_2} \min_{\mathbf{u} \in \mathcal{K}_1} M(\mathbf{u}, \mathbf{v}) \quad \textbf{(Goal of } \mathcal{P}_2\textbf{)} \tag{4}$$

This definition of a game makes sense if there exists a point $(\mathbf{u}^*, \mathbf{v}^*)$, such that neither $\mathcal{P}_1$ nor $\mathcal{P}_2$ may increase their utility by unilateral deviation. Such a point $(\mathbf{u}^*, \mathbf{v}^*)$ is called a *Pure Nash Equilibrium*, which is formally defined as a point which satisfies the following conditions:

$$M(\mathbf{u}^*, \mathbf{v}^*) \leq \min_{\mathbf{u} \in \mathcal{K}_1} M(\mathbf{u}, \mathbf{v}^*), \quad \& \quad M(\mathbf{u}^*, \mathbf{v}^*) \geq \max_{\mathbf{v} \in \mathcal{K}_2} M(\mathbf{u}^*, \mathbf{v}) .$$

While a pure Nash equilibrium does not always exist, the pioneering work of Nash et al. (1950) established that there always exists a *Mixed Nash Equilibrium* (MNE or simply equilibrium), i.e., there always exist two distributions $\mathcal{D}_1, \mathcal{D}_2$ such that,

$$\mathbb{E}_{(\mathbf{u},\mathbf{v}) \sim \mathcal{D}_1 \times \mathcal{D}_2}[M(\mathbf{u}, \mathbf{v})] \leq \min_{\mathbf{u} \in \mathcal{K}_1} \mathbb{E}_{\mathbf{v} \sim \mathcal{D}_2}[M(\mathbf{u}, \mathbf{v})], \; \& \; \mathbb{E}_{(\mathbf{u},\mathbf{v}) \sim \mathcal{D}_1 \times \mathcal{D}_2}[M(\mathbf{u}, \mathbf{v})] \geq \max_{\mathbf{v} \in \mathcal{K}_2} \mathbb{E}_{\mathbf{u} \sim \mathcal{D}_1}[M(\mathbf{u}, \mathbf{v})] .$$

Finding an exact MNE might be computationally hard, and we are usually satisfied with finding an approximate MNE. This is defined below,

**Definition 1.** *Let $\varepsilon > 0$. Two distributions $\mathcal{D}_1, \mathcal{D}_2$ are called $\varepsilon$-MNE if the following holds,*

$$\mathbb{E}_{(\mathbf{u},\mathbf{v}) \sim \mathcal{D}_1 \times \mathcal{D}_2}[M(\mathbf{u}, \mathbf{v})] \leq \min_{\mathbf{u} \in \mathcal{K}_1} \mathbb{E}_{\mathbf{v} \sim \mathcal{D}_2}[M(\mathbf{u}, \mathbf{v})] + \varepsilon,$$

$$\mathbb{E}_{(\mathbf{u},\mathbf{v}) \sim \mathcal{D}_1 \times \mathcal{D}_2}[M(\mathbf{u}, \mathbf{v})] \geq \max_{\mathbf{v} \in \mathcal{K}_2} \mathbb{E}_{\mathbf{u} \sim \mathcal{D}_1}[M(\mathbf{u}, \mathbf{v})] - \varepsilon .$$

**Terminology:** In the sequel when we discuss zero-sum games, we shall sometimes use the GAN terminology, relating the $\min$ player $\mathcal{P}_1$ as the *generator*, and the $\max$ player $\mathcal{P}_2$, as the *discriminator*.

**No-Regret & Zero-sum Games:** In zero-sum games, no-regret algorithms may be used to find an approximate MNE. Unfortunately, computationally tractable no-regret algorithms do not always exist. An exception is the setting when $M$ is convex-concave. In this case, the players may invoke the powerful no-regret methods from online convex optimization to (approximately) solve the game. This seminal idea was introduced in Freund & Schapire (1999), where it was demonstrated how to invoke no-regret algorithms during $T$ rounds to obtain an approximation guarantee of $\varepsilon = O(1/\sqrt{T})$ in zero-sum matrix games. This was later improved by Daskalakis et al. (2015), and Rakhlin & Sridharan (2013), demonstrating a guarantee of $\varepsilon = O(\log T/T)$. The result that we are about to present builds on the scheme of Freund & Schapire (1999).

---

[4]Tikhonov regularization $\mathcal{R}(\mathbf{u}) = \|\mathbf{u}\|^2$ is one of the most popular regularizers.

## 3 FINDING EQUILIBRIUM IN GANS

**Why Mixed Equilibrium?** In this work, our ultimate goal is to efficiently find an approximate MNE for the game. However, in GANs, we are usually interested in designing good generators, and one might ask whether finding an equilibrium serves this cause better than solving the minimax problem, i.e., finding $\mathbf{u} \in \mathrm{argmin}_{\mathbf{u} \in \mathcal{K}_1} \max_{\mathbf{v} \in \mathcal{K}_2} M(\mathbf{u}, \mathbf{v})$. Interestingly, the minimax value of the equilibrium generator is always smaller than the minimax value of any pure strategy. Actually, the equilibrium strategy of the generator might be much better. This benefit of finding an equilibrium can be demonstrated on the following simple zero-sum game. Consider the following *paper-rock-scissors* game, i.e. a zero-sum game with the minimax objective

$$\min_{i \in \{1,2,3\}} \max_{j \in \{1,2,3\}} M(i,j) \;; \text{ where } M = \begin{bmatrix} 0 & -1 & 1 \\ 1 & 0 & -1 \\ -1 & 1 & 0 \end{bmatrix}.$$

Solving for the minimax objective yields a pure strategy with a minimax value of 1; conversely, the equilibrium strategy of the $\min$ player is a uniform distribution over actions; and its minimax value is 0. Thus, finding an equilibrium by allowing mixed strategies implies a smaller minimax value, and as we show in the Section 3.3 this is true in general. In the context of GANs, this result means that the mixed generator provided by the equilibrium solution will "fool" any adversary at least as good as any single generator. Similarly, the mixed discriminator provided by the equilibrium solution will discern any generator at least as good as any single discriminator.

The rest of this section presents a method that efficiently finds an equilibrium for semi-shallow GANs (see Fig. 1(b)). Such architectures do not induce a convex-concave game, and therefore the result of Freund & Schapire (1999) does not directly apply. Nevertheless, we show that semi-shallow GANs imply a game structure which gives rise to an efficient procedure for finding an equilibrium. In Sec. 3.1 we show that semi-shallow GANs define games with a property that we denote as semi-concave. Later, Sec. 3.2 provides an algorithm with provable guarantees for such games. Finally, in Section 3.3 we show that the minimax objective of the generator's equilibrium strategy is optimal with respect to the minimax objective.

### 3.1 SEMI-SHALLOW GANS

Semi-shallow GANs do not lead to a convex-concave game. Nonetheless, here we show that for an appropriate choice of the activation function they induce a game that is concave with respect to the discriminator. Later, in Sec. 3.2, we show that this property allows to efficiently find an equilibrium.

**Proposition 1.** *Consider the GAN objective in Eq.* (1) *and assume that the discriminator is a single-layer network with a sigmoid activation function, meaning* $h_{\mathbf{v}}(\mathbf{x}) = 1/(1 + \exp(-\mathbf{v}^\top \mathbf{x}))$, *where* $\mathbf{v} \in \mathbb{R}^n$. *Then the GAN objective is concave in* $\mathbf{v}$.

Note that the above is not restricted to the sigmoid activation function, and it also holds for other choices of the activation function [5].

### 3.2 SEMI-CONCAVE ZERO-SUM GAMES

Here we discuss the setting of zero-sum games (see Eq. (4)) which are *semi-concave*. Formally a game, $M$, is semi-concave if for any fixed $\mathbf{u}_0 \in \mathcal{K}_1$ the function $g(\mathbf{v}) := M(\mathbf{u}_0, \mathbf{v})$ is concave in $\mathbf{v}$. Algorithm 1 presents our method for semi-concave games. This algorithm is an instantiation of the scheme derived by Freund & Schapire (1999), with specific choices of the online algorithms $\mathcal{A}_1, \mathcal{A}_2$, used by the players. Note that both $\mathcal{A}_1, \mathcal{A}_2$ are two different instances of the FTRL approach presented in Eq. (3).

Let us discuss Algorithm 1 and then present its guarantees. First note that each player calculates a sequence of $T$ points based on an online algorithm $\mathcal{A}_1/\mathcal{A}_2$. Interestingly, the sequence of (loss/reward) functions given to the online algorithm is based on the game objective $M$, and

---

[5]As an example the proposition holds by choosing $h_{\mathbf{v}}(\mathbf{x})$ to be the cumulative gaussian distribution, i.e. $h_{\mathbf{v}}(\mathbf{x}) = \Phi(\mathbf{v}^\top \mathbf{x})$, where $\Phi(a) = \int_{y=-\infty}^{a} (2\pi)^{-0.5} \exp(-y^2/2) dy$ . Note that the logarithm of $h_{\mathbf{v}}(x)$ for the sigmoid and cumulative gaussian activations correspond to the well known logit and probit models, McCullagh & Nelder (1989).

---

**Algorithm 1** CHEKHOV GAN

---

**Input**: #steps $T$, Game objective $M(\cdot, \cdot)$
**for** $t = 1 \ldots T$ **do**
   Calculate:

$$(\text{Alg. } \mathcal{A}_1) \quad \mathbf{u}_t \leftarrow \underset{\mathbf{u} \in \mathcal{K}_1}{\arg\min} \sum_{\tau=0}^{t-1} f_\tau(\mathbf{u}) \quad \& \quad (\text{Alg. } \mathcal{A}_2) \quad \mathbf{v}_t \leftarrow \underset{\mathbf{v} \in \mathcal{K}_2}{\arg\max} \sum_{\tau=0}^{t-1} \nabla g_\tau(\mathbf{v}_\tau)^\top \mathbf{v} - \frac{\sqrt{T}}{2\eta_0} \|\mathbf{v}\|^2$$

   Update: $\qquad\qquad\qquad f_t(\cdot) = M(\cdot, \mathbf{v}_t) \quad \& \quad g_t(\cdot) = M(\mathbf{u}_t, \cdot)$
**end for**
**Output mixed strategies**: $\mathcal{D}_1 \sim \text{Uni}\{\mathbf{u}_1, \ldots, \mathbf{u}_T\}$, $\mathcal{D}_2 \sim \text{Uni}\{\mathbf{v}_1, \ldots, \mathbf{v}_T\}$.

---

also on the decisions made by the other player. For example, the loss sequence that $\mathcal{P}_1$ receives is $\{f_t(\mathbf{u}) := M(\mathbf{u}, \mathbf{v}_t)\}_{t \in [T]}$. After $T$ rounds we end up with two mixed strategies $\mathcal{D}_1, \mathcal{D}_2$, each being a uniform distribution over the respective online decisions $\{\mathbf{u}_t\}_{t \in [T]}, \{\mathbf{v}_t\}_{t \in [T]}$. Note that the first decision points $\mathbf{u}_1, \mathbf{v}_1$ are set by $\mathcal{A}_1, \mathcal{A}_2$ before encountering any (loss/reward) function, and the dummy functions $f_0(\mathbf{u}) = 0, g_0(\mathbf{v}) = 0$ are only introduced in order to simplify the exposition. Since $\mathcal{P}_1$'s goal is to minimize, it is natural to think of the $f_t$'s as loss functions, and measure the guarantees of $\mathcal{A}_1$ according to the regret as defined in Equation (2). Analogously, since $\mathcal{P}_2$'s goal is to maximize, it is natural to think of the $g_t$'s as reward functions, and measure the guarantees of $\mathcal{A}_2$ according to the following appropriate definition of regret, $\text{Regret}_T^{\mathcal{A}_2} = \max_{\mathbf{v}^* \in \mathcal{K}_2} \sum_{t=1}^T g_t(\mathbf{v}^*) - \sum_{t=1}^T g_t(\mathbf{v}_t)$.

The following theorem presents our guarantees for semi-concave games:

**Theorem 1.** *Let $\mathcal{K}_2$ be a convex set. Also, let $M$ be a semi-concave zero-sum game, and assume $M$ is $L$-Lipschitz continuous. Then upon invoking Alg. 1 for $T$ steps it outputs mixed strategies $(\mathcal{D}_1, \mathcal{D}_2)$ that are $\varepsilon$-MNE, where $\varepsilon = O(1/\sqrt{T})$.*

The most important point to note is that the accuracy of the approximation $\varepsilon$ improves as the number of iterations $T$ grows. This lets us obtain an arbitrarily good approximation for a large enough $T$. As mentioned before, both $\mathcal{A}_1, \mathcal{A}_2$ are two different instances of the FTRL approach presented in Eq. (3). Concretely, Alg. $\mathcal{A}_1$ is in fact follow-the-leader (FTL), i.e., FTRL without regularization. Alg. $\mathcal{A}_2$ also uses the FTRL scheme. Yet, instead of the original reward functions, $g_t(\cdot)$, it utilizes linear approximations $\tilde{g}_t(\mathbf{v}) = \nabla g_t(\mathbf{v}_t)^\top \mathbf{v}$. Also note the use of the (minus) square $\ell_2$ norm as regularization[6]. The $\eta_0$ parameter depends on the Lipschitz constant of $M$ as well as on the diameter of $\mathcal{K}_2$ defined as, $d_2 := \max_{\mathbf{v}_1, \mathbf{v}_2 \in \mathcal{K}_2} \|\mathbf{v}_1 - \mathbf{v}_2\|$. Concretely, $\eta_0 = d_2/\sqrt{2}L$.

Next we provide a short proof sketch for Thm. 1. The full proof appears in Appendix A.

*Proof sketch.* The proof makes use of a theorem due to Freund & Schapire (1999) which shows that if both $\mathcal{A}_1$ and $\mathcal{A}_2$ ensure no-regret then it implies convergence to an approximate MNE. Since the game is concave with respect to $\mathcal{P}_2$, it is well known that the FTRL version $\mathcal{A}_2$ appearing in Thm. 1 is a no-regret strategy (see e.g. Hazan et al. (2016)). The challenge is therefore to show that $\mathcal{A}_1$ is also a no-regret strategy. This is non-trivial, especially for semi-concave games that do not necessarily have any special structure with respect to the generator [7]. However, the loss sequence received by the generator is not arbitrary, but rather it follows a special sequence based on the choices of the discriminator, $\{f_t(\cdot) = M(\cdot, \mathbf{v}_t)\}_t$. In the case of semi-concave games, the sequence of discriminator decisions, $\{\mathbf{v}_t\}_t$ has a special property which "stabilizes" the loss sequence $\{f_t\}_t$, which in turn enables us to establish no-regret for $\mathcal{A}_1$. □

**Remark:** Note that Alg. $\mathcal{A}_1$ in Thm. 1 assumes the availability of an oracle that can efficiently find a global minimum for the FTL objective, $\sum_{\tau=0}^{t-1} f_\tau(\mathbf{u})$. This involves a minimization over a sum of generative networks. Therefore, our result may be seen as a reduction from the problem of finding

---

[6]Note the use of the minus sign in the regularization since the discriminator's goal is to maximize, thus the $g_t(\cdot)$, may be thought of as reward functions.

[7]The result of Hazan & Koren (2016) shows that there does not exist any efficient no-regret algorithm, $\mathcal{A}_1$, in the general case where the loss sequence $\{f_t(\cdot)\}_{t \in [T]}$ received by $\mathcal{A}_1$ is arbitrary.

an equilibrium to an offline optimization problem. This reduction is not trivial, especially in light of the negative results of Hazan & Koren (2016), which imply that in the general case finding an equilibrium is hard, even with such an efficient offline optimization oracle at hand. Thus, our result enables to take advantage of progress made in supervised deep learning in order to efficiently find an equilibrium for GANs.

### 3.3 MINIMAX VALUE OF EQUILIBRIUM STRATEGY

In GANs we are mainly interested in ensuring the performance of the generator (resp. discriminator) with respect to the *minimax* (resp. *maximin*) objective. Let $(\mathcal{D}_1, \mathcal{D}_2)$ be the pair of mixed strategies that Algorithm 1 outputs. Note that the minimax value of $\mathcal{D}_1$ might be considerably smaller than the pure minimax value, as is shown in the example regarding the *paper-rock-scissors* game (see Sec. 3). The next lemma shows that the mixed strategy $\mathcal{D}_1$ is always (approximately) better with respect to the pure minimax value (see proof in appendix B.2)

**Lemma 1.** *The mixed strategy $\mathcal{D}_1$ that Algorithm 1 outputs is $\varepsilon$-optimal with respect to the minimax value, i.e.,*

$$\max_{\mathbf{v} \in \mathcal{K}_2} \mathbb{E}_{\mathbf{u} \sim \mathcal{D}_1}[M(\mathbf{u}, \mathbf{v})] \leq \min_{\mathbf{u} \in \mathcal{K}_1} \max_{\mathbf{v} \in \mathcal{K}_2} M(\mathbf{u}, \mathbf{v}) + \varepsilon$$

*where $\varepsilon$ here is equal to the one defined in Thm. 2.*

Analogous result hold for $\mathcal{D}_2$ with respect to the pure maximin objective.

## 4 PRACTICAL CHEKHOV GAN ALGORITHM FOR DEEP ARCHITECTURES

---

**Algorithm 2** Practical CHEKHOV GAN

**Input**: #steps $T$, Game objective $M(\cdot, \cdot)$, number of past states $K$, spacing $m$
**Initialize**: Set loss/reward $f_0(\cdot) = 0, g_0(\cdot) = 0$, initialize queues $\mathcal{Q}_1$.insert($f_0$), $\mathcal{Q}_2$.insert($g_0$)
**for** $t = 1 \ldots T$ **do**
    Update generator and discriminator based on a mini-batch of noise samples and data samples:

$$\mathbf{u}_{t+1} \leftarrow \mathbf{u}_t - \eta_t \cdot \nabla_{\mathbf{u}_t} \left( \frac{1}{|\mathcal{Q}_1|} \sum_{f \in \mathcal{Q}_1} f(\mathbf{u}) + \frac{C}{\sqrt{t}} \|\mathbf{u}\|^2 \right) \ \& \ \mathbf{v}_{t+1} \leftarrow \mathbf{v}_t - \eta_t \cdot \nabla_{\mathbf{v}_t} \left( \frac{1}{|\mathcal{Q}_2|} \sum_{g \in \mathcal{Q}_2} g(\mathbf{v}) - \frac{C}{\sqrt{t}} \|\mathbf{v}\|^2 \right)$$

    Calculate: $f_t(\cdot) = M(\cdot, \mathbf{v}_t) \ \& \ g_t(\cdot) = M(\mathbf{u}_t, \cdot)$
    Update $\mathcal{Q}_1$ and $\mathcal{Q}_2$ (see main text or Algorithm 3 in the appendix)
**end for**
**Output mixed strategies**: $\mathcal{D}_1 \sim \text{Uni}\{\mathbf{u}_1, \ldots, \mathbf{u}_K \in \mathcal{Q}_1\}$, $\mathcal{D}_2 \sim \text{Uni}\{\mathbf{v}_1, \ldots, \mathbf{v}_K \in \mathcal{Q}_2\}$.

---

In Section 3 we described a method (Alg. 1) which provably reaches an equilibrium for semi-shallow GANs. This method considers the whole history of generators and discriminators in making a decision at each round, which contrasts with the standard GAN training method that only considers the last generator and discriminator. Another difference is that our method outputs a mixed model (i.e., generator and discriminator) rather than a single model.

Building on the theoretical approach introduced in Section 3, we now present a practical method (Alg. 2) which can be efficiently applied to train common *deep* GAN architectures. Algorithm 2 combines the ideas of **(a)** considering the history of generators and discriminators at each update, and **(b)** outputting a mixed strategy, while only requiring an access to gradient information which can be efficiently obtained by running back-propagation. Next we discuss Alg. 2 in more details and highlight the differences compared to the theoretical approach:

**(i)** We use the FTRL objective (Eq. (3)) for both players. Note that Alg. $\mathcal{A}_1$ appearing in Thm 1 uses FTRL with linear approximations, which is only appropriate for semi-concave games.

**(ii)** As calculating the global minimizer of the FTRL objective is impractical, we instead update the weights based on the gradients of the FTRL objective. This can be done by using traditional optimization techniques such as SGD or Adam. Thus the update at each round depends on the

gradients of the *past* generators and discriminators. This differs from the standard GAN training which only employs the gradient of the *last* generator and discriminator.

**(iii)** The full FTRL algorithm requires saving the entire history of past generators/discriminators, which is computationally intractable. We find it sufficient to maintain a summary of the history using a small number of representative models. In order to capture a diverse subset of the history, we keep a queue $\mathcal{Q}$ containing $K := |\mathcal{Q}|$ states (models). The spacing between consecutive models is determined by the following heuristic: every $m$ update steps we remove the oldest model in the queue and add the current one. The number of steps between switches, $m$, can be set as a constant, but our experiments revealed it is more effective to keep $m$ small at the beginning and increase its value as the number of rounds increases. We hypothesize that as the training progresses and the individual models become more discriminative, we should switch the models at a lower rate, keeping them more spaced out. The pseudo-code and a detailed description of the algorithm appears in the Appendix.

**Intuition.** In practice GANs commonly exhibit a non-convergent behavior. As a consequence, the generator oscillates between generating different modes from the target distribution. This is hypothesized to be due to the differences of the minimax and maximin solutions of the game (Goodfellow, 2016). If the order of the min and max operations switch, the minimization with respect to the generator's parameters is performed in the inner loop. This causes the generator to map every latent code to one or very few points for which the discriminator believes are likely. As simultaneous gradient descent updates do not clearly prioritize any specific ordering of minimax or maximin, in practice we often obtain results that resemble the latter.

In contrast, CHEKHOV GAN takes advantage of the history of the player's actions which yields better gradient information. Intuitively, the generator is updated such that it fools the past discriminators. In order to do so, the generator has to spread its mass more fairly according to the true data distribution. The discriminator can no longer simply learn to put low probability on the few modes of generated samples, which causes oscillations.

The mode collapse problems of GANs is also closely related to the phenomenon of catastrophic forgetting (Seff et al., 2017). When GANs are trained sequentially on samples coming from different modes, the discriminator tends to forget the previous modes it has learned about. This leads to having generated samples that focus only on the last or most prominent modes. By introducing a history of samples from previous generators, the discriminator is less likely to forget the part of the space that it has already learned.

Fig. 2 illustrates the mode collapse problem. The data consists of a mixture of 7 Gaussians with different sampling probabilities whose centers are aligned in a circle. As two modes have higher probabilities and are seen more frequently, they attract the gradients towards them and cause mode collapse and forgetting. Chekhov GAN manages to recover the true data distribution in this case as well, unlike vanilla GANs.

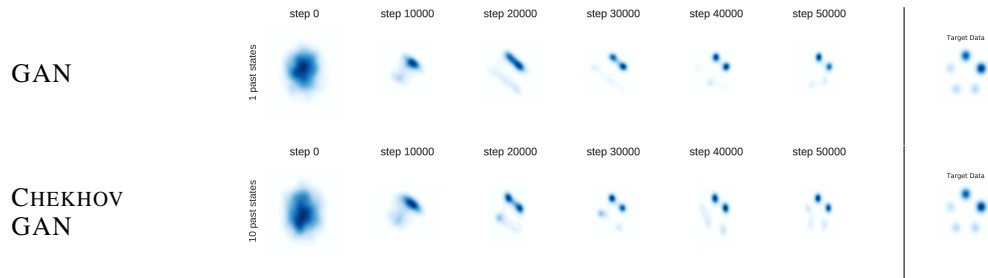

Figure 2: Mode Collapse on a Gaussian Mixture. We show heat maps of the generator distribution over time, as well as the target data distribution in the last column. Standard GAN updates (top row) cause mode collapse, whereas CHEKHOV GAN using $K = 10$ past steps (bottom row) spreads its mass over all the modes of the target distribution.

## 5    Experimental results

We now demonstrate that CHEKHOV GAN yields improved stability and sample diversity. To do so, we use a comparable number of datasets and baselines as standard GAN approaches, e.g. Metz et al. (2016); Arjovsky & Bottou (2017). We test our method on models where the traditional GAN training has difficulties converging and engages in a behavior of mode collapse. We also perform experiments on harder tasks using the DCGAN architecture (Radford et al., 2015) which is commonly used in the literature. Note that the DCGAN architecture, when trained using standard techniques, still suffers from instabilities and mode collapse (Nagarajan & Kolter, 2017; Roth et al., 2017). We here demonstrate that CHEKHOV GAN reduces mode dropping while retaining high visual sample quality. For all of the experiments, we generate from the newest generator only. Experimental details and comparisons to additional baselines, as well as a set of recommended hyperparameters are available in Appendix D and Appendix C, respectively.

### 5.1    Non-convergence and Mode Dropping

#### 5.1.1    Augmented MNIST

We first evaluate the ability of our approach to avoid mode collapse on real image data coming from an augmented version of the MNIST dataset. Similarly to (Metz et al., 2016; Che et al., 2016), we combine three randomly selected MNIST digits to form 3-channel images, resulting in a dataset with 1000 different classes, one for each of the possible combinations of the ten MNIST digits.

We train a simplified DCGAN architecture (see details in Appendix D) with both GAN and CHEKHOV GAN with a different number of saved past states. The evaluation of each model is done as follows. We generate a fixed amount of samples (25,600) from each model and classify them using a pre-trained MNIST classifier with an accuracy of $99.99\%$. The models that exhibit less mode collapse are expected to generate samples from most of the 1000 modes.

We report two different evaluation metrics in Table 1: i) the number of classes for which a model generated at least one sample, and ii) the reverse KL divergence. The reverse KL divergence between the model and the target data distribution is computed by considering that the data distribution is a uniform distribution over all classes.

| Models | 0 states (GAN) | 5 states | 10 states |
|---|---|---|---|
| Generated Classes | $629 \pm 121.08$ | $743 \pm 64.31$ | $795 \pm 37$ |
| Reverse KL | $1.96 \pm 0.64$ | $1.40 \pm 0.21$ | $1.24 \pm 0.17$ |

Table 1: Stacked MNIST: Number of generated classes out of 1000 possible combinations, and the reverse KL divergence score. The results are averaged over 10 runs.

### 5.2    Image Modeling

We turn to the evaluation of our model for the task of generating rich image data for which the modes of the data distribution are unknown. In the following, we perform experiments that indirectly measure mode coverage through metrics based on the sample diversity and quality.

#### 5.2.1    Inference via Optimization on CIFAR10

We train a DCGAN architecture on CIFAR10 (Krizhevsky & Hinton, 2009) and evaluate the performance of each model using the inference via optimization technique introduced in (Metz et al., 2016) and explained in Appendix D.3.3.

The average MSE over 10 rounds using different seeds is reported in Table 2. Using CHEKHOV GAN with as few as 5 past states results in a significant gain which can be further improved by increasing the number of past states to 10 and 25. In addition, the training procedure becomes more stable as indicated by the decrease in the standard deviation. The percentage of mini-batches that achieve the lowest reconstruction loss with the different models is given in Table 2. This can also be visualized by comparing the closest images $x_{closest}$ from each model to real target images $x_{target}$ as

shown in Figure 3. The images are randomly selected images from the batch which has the largest absolute difference in MSE between GAN and CHEKHOV GAN with 25 states. The samples obtained by the original GAN are often blurry while samples from CHEKHOV GAN are both sharper and exhibit more variety, suggesting a better coverage of the true data distribution.

| Target | Past States | 0 (GAN) | 5 states | 10 states | 25 states |
|--------|-------------|---------|----------|-----------|-----------|
| Train | MSE | $61.13 \pm 3.99$ | $58.84 \pm 3.67$ | $56.99 \pm 3.49$ | **$48.42 \pm 2.99$** |
| Set | Best Rank (%) | 0 % | 0 % | 18.66 % | **81.33 %** |
| Test | MSE | $59.5 \pm 3.65$ | $56.66 \pm 3.60$ | $53.75 \pm 3.47$ | **$46.82 \pm 2.96$** |
| Set | Best Rank (%) | 0 % | 0 % | 17.57 % | **82.43 %** |

Table 2: CIFAR10: MSE between target images from the train and test set and the best rank which consists of the percentage of minibatches containing target images that can be reconstructed with the lowest loss across various models. We use 20 different minibatches, each containing 64 target images. Increasing the number of past states for CHEKHOV GAN allows the model to better match the real images.

| Real Image | 0 states | 10 states | 25 states |
|------------|----------|-----------|-----------|
| | 4.14 | 2.37 | 2.19 |
| | 4.17 | 2.97 | 2.58 |

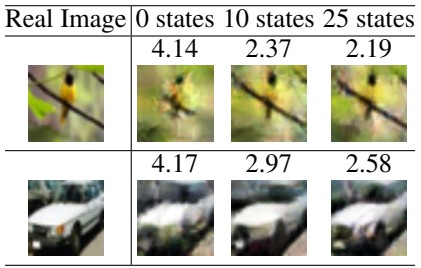

| Real Image | 0 states | 10 states | 25 states |
|------------|----------|-----------|-----------|
| | 3.06 | 2.51 | 2.35 |
| | 1.12 | 0.30 | 0.39 |

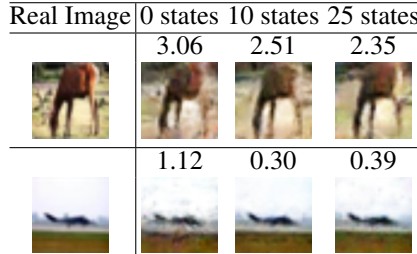

Table 3: CIFAR10: Target images from the test set are shown on the left. The images from each model that best resemble the target image are shown for different number of past states: 0 (GAN), 10 and 25 (CHEKHOV GAN ). The reconstruction MSE loss is indicated above each image.

Note that the numbers quoted in our paper can directly be compared to the ones reported in unrolled GAN (Metz et al., 2016) since we have used the same architecture and choice of hyper-parameters. We include a comparison in the appendix.

### 5.2.2    ESTIMATION OF MISSING MODES ON CELEBA

We estimate the number of missing modes on the CelebA dataset (Liu et al., 2015) by using an auxiliary discriminator as performed in (Che et al., 2016). The experiment consists of two phases. In the first phase we train GAN and CHEKHOV GAN models and generate a fixed number of images. In the second phase we independently train a noisy discriminator using the DCGAN architecture where the training data is the previously generated data from each of the models, respectively. The noisy discriminator is then used as a mode estimator. The test images from CelebA that are classified as fake by the mode estimator are considered as images belonging to a missing mode. Table 4 shows the number of missed modes for the two models. Generated samples from each model are given in the Appendix.

| $\sigma$ | 0 states (GAN) | 5 states (CHEKHOV GAN ) |
|----------|----------------|--------------------------|
| 0.25 | $3004 \pm 4154$ | $1407 \pm 1848$ |
| 0.5 | $2568.25 \pm 4148$ | $1007 \pm 1805$ |

Table 4: CelebA: Number of images from the test set that the auxiliary discriminator classifies as not real. Gaussian noise with variance $\sigma^2$ is added to the input of the auxiliary discriminator, with the standard deviation shown in the first row. The test set consists of 50,000 images.

Interestingly, even with small number of past states (K=5), CHEKHOV GAN manages to stabilize the training and generate more diverse samples on all the datasets. In terms of computational complexity, our algorithm scales linearly with $K$. However, all the elements in the sum are independent and can be computed efficiently in a parallel manner.

| Model | Inception Score | Fréchet Inception Distance |
|---|---|---|
| GAN | $5.89 \pm 0.15$ | $42.99 \pm 6.85$ |
| Chekhov GAN | $6.02 \pm 0.15$ | $40.97 \pm 1.03$ |
| Unrolled GAN | $5.51 \pm 0.13$ | $53.83 \pm 6.09$ |
| MIX+GAN | $6.03 \pm 0.16$ | $41.79 \pm 5.10$ |
| WGAN | $4.85 \pm 0.12$ | $66.59 \pm 1.28$ |
| Chekhov WGAN | $5.13 \pm 0.13$ | $58.82 \pm 1.34$ |
| MIX+WGAN | $4.91 \pm 0.12$ | $64.89 \pm 4.53$ |
| WGAN GP | $5.31 \pm 0.13$ | $52.52 \pm 5.28$ |
| Real Data | $11.24 \pm 0.12$ | $5.19 \pm 0.02$ |

Table 5: Inception score (higher is better) and FID (lower is better) on Cifar10: The first section shows GAN variants and the second shows WGAN variants. The results are averaged over 8 runs.

## 5.3 INCEPTION SCORE AND FRÉCHET INCEPTION DISTANCE

We compare our algorithm against several state-of-the-art GAN methods using the Inception Score (Salimans et al., 2016) as metric, as well as its improved version, the Fréchet Inception Distance (Heusel et al., 2017). All the models used in this experiment rely on the same architectures for both the generator and the discriminator. We generated 50,000 samples for the calculation of the two metrics. Table 5 shows the results on Cifar10 when all models are trained in an unsupervised fashion.

We apply our algorithm on top of GAN and WGAN (Arjovsky et al., 2017), as both define a minimax game. Both Chekhov GAN and Chekhov WGAN are trained using 5 past states. For fair comparison, MIX+GAN and MIX+WGAN (Arora et al., 2017) use a mixture of 5 generators and discriminators and the number of unrolling steps for Unrolled GAN (Metz et al., 2016) is set to 5. The number of parameters for MIX+(GAN/WGAN) is 5 times the number of parameters for all the other baselines.

We find that by applying our algorithm on top of WGAN, we get an improvement of 5.13 vs. 4.85 (inception score) and 58.82 vs. 66.59 (FID). Throughout the training, Chekhov WGAN consistently outperforms WGAN by achieving higher scores and lower distances, as shown in Figure 3. Across the WGAN-based variants, Chekhov WGAN consistently outperforms MIX+WGAN as well, while WGAN GP (Gulrajani et al., 2017) achieves the best scores within the group. However, as shown in Figure 4, Chekhov WGAN improves upon WGAN and reaches almost the same level of performance as WGAN GP.

In terms of the GAN-based variants, Chekhov GAN consistently outperforms Unrolled GAN by a large margin across all epochs. MIX+GAN, GAN and Chekhov GAN achieve comparable scores in terms of both metrics, but Chekhov GAN does so while reducing the variance significantly (see Figure 4).

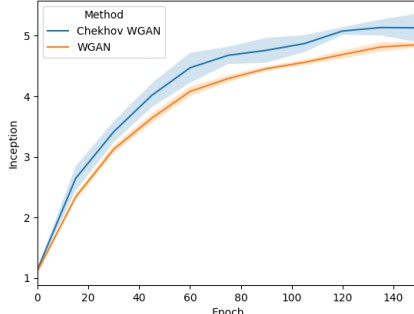 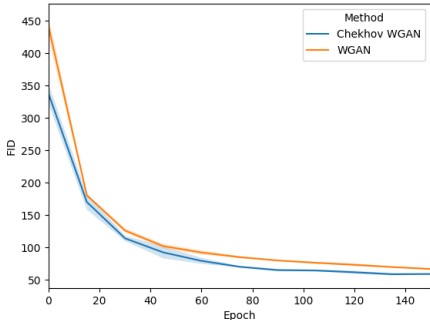

Figure 3: Chekhov GAN consistently outperforms WGAN across both metrics: (left) Inception score, (right) Fréchet Inception Distance (FID). The shaded area denotes the standard deviation.

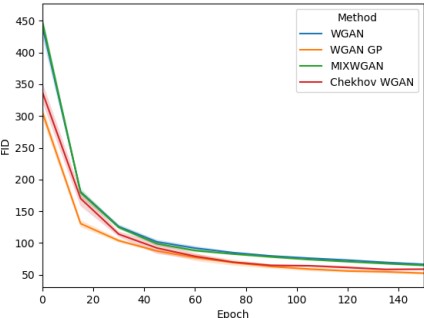 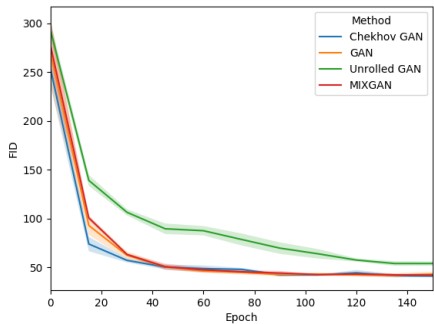

Figure 4: Comparison of FID: (left) WGAN-based variants, (right) GAN-based variants. The shaded area denotes the standard deviation.

## 6 CONCLUSION

We have presented a principled approach to training GANs, which is guaranteed to reach convergence to a mixed equilibrium for semi-shallow architectures. Empirically, our approach presents several advantages when applied to commonly used GAN architectures, such as improved stability or reduction in mode dropping. Our results open an avenue for the use of online-learning and game-theoretic techniques in the context of training GANs. One question that remains open is whether the theoretical guarantees can be extended to more complex architectures.

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

# A  ANALYSIS

Here we provide the proof of Thm. 1.

*Proof.* We make a use of a theorem due to Freund & Schapire (1999) which shows that if both $\mathcal{A}_1, \mathcal{A}_2$ ensure no regret implies approximate MNE. For completeness we provide its proof in Sec. A.2.

**Theorem 2.** *The mixed strategies* $(\mathcal{D}_1, \mathcal{D}_2)$ *that Algorithm 1 outputs are* $\varepsilon$-*MNE, where*

$$\varepsilon := \left( B_T^{\mathcal{A}_1} + B_T^{\mathcal{A}_2} \right) /T .$$

*here* $B_T^{\mathcal{A}_1}, B_T^{\mathcal{A}_2}$ *are bounds on the regret of* $\mathcal{A}_1, \mathcal{A}_2$.

According to Thm. 2, it is sufficient to show that both $\mathcal{A}_1$, and $\mathcal{A}_2$ ensure a regret bound of $O(\sqrt{T})$.

**Guarantees for** $\mathcal{A}_2$**:** this FTRL version is well known in online learning, and its regret guarantees can be found in the literature, (e.g, Theorem 5.1 in  Hazan et al. (2016)). The following lemma provides its guarantees,

**Lemma 2.** *Let* $d_2$ *be the diameter of* $\mathcal{K}_2$. *Invoking* $\mathcal{A}_2$ *with* $\eta_0 = d_2/\sqrt{2}L$, *ensures the following regret bound over the sequence of concave functions* $\{g_t\}_{t \in [T]}$,

$$Regret_T^{\mathcal{A}_2}(g_1, \ldots, g_T) \leq Ld_2\sqrt{2T} .$$

*Moreover, the following applies for the sequence* $\{\mathbf{v}_t\}_{t \in [T]}$ *generated by* $\mathcal{A}_2$,

$$\|\mathbf{v}_{t+1} - \mathbf{v}_t\| \leq d_2/\sqrt{2T} .$$

Note that the proof heavily relies on the concavity of the $g_t(\cdot)$'s, which is due to the concavity of the game with respect to the discriminator. For completeness we provide a proof of the second part of the lemma in Sec. B.3.

**Guarantees for** $\mathcal{A}_1$**:** By Lemma 2, the sequence generated by the discriminator not only ensures low regret but is also stable in the sense that consecutive decision points are close by. This is the key property which will enable us to establish a regret bound for algorithm $\mathcal{A}_1$. Next we state the guarantees of $\mathcal{A}_1$,

**Lemma 3.** *Let* $C := \max_{\mathbf{u} \in \mathcal{K}_1, \mathbf{v} \in \mathcal{K}_2} |M(\mathbf{u}, \mathbf{v})|$. *Consider the loss sequence appearing in Alg 1,* $\{f_t(\cdot) := M(\cdot, \mathbf{v}_t)\}_{t \in [T]}$. *Then algorithm* $\mathcal{A}_1$ *ensures the following regret bound over this sequence,*

$$Regret_T^{\mathcal{A}_1}(f_1, \ldots, f_T) \leq \frac{1}{\sqrt{2}}Ld_2\sqrt{T} + 2C .$$

Combining the regret bounds of Lemmas 2, 3 and Theorem 2 concludes the proof of Thm. 1.  □

## A.1  PROOF OF LEMMA 3

*Proof.* We use the following regret bound regarding the FTL (follow-the-leader) decision rule, derived in Kalai & Vempala (2005) (see also Shalev-Shwartz et al. (2012)),

**Lemma 4.** *For any sequence of loss functions* $\{f_t\}_{t \in [T]}$, *the regret of FTL is bounded as follows,*

$$Regret_T^{FTL}(f_1, \ldots, f_T) \leq \sum_{t=1}^{T} \left( f_t(\mathbf{u}_t) - f_t(\mathbf{u}_{t+1}) \right) .$$

Since $\mathcal{A}_1$ is FTL, the above bound applies. Thus, using the above bound together with the stability of the $\{\mathbf{v}_t\}_{t\in[T]}$ sequence we obtain,

$$
\begin{aligned}
\text{Regret}_T^{\mathcal{A}_1} &\leq \sum_{t=1}^{T} \big(f_t(\mathbf{u}_t) - f_t(\mathbf{u}_{t+1})\big) \\
&= \sum_{t=1}^{T-1} \big(f_t(\mathbf{u}_t) - f_t(\mathbf{u}_{t+1}) + f_{t+1}(\mathbf{u}_{t+1}) - f_{t+1}(\mathbf{u}_{t+1})\big) + \big(f_T(\mathbf{u}_T) - f_T(\mathbf{u}_{T+1})\big) \\
&= \sum_{t=1}^{T-1} \big(f_{t+1}(\mathbf{u}_{t+1}) - f_t(\mathbf{u}_{t+1})\big) + \sum_{t=1}^{T-1} \big(f_t(\mathbf{u}_t) - f_{t+1}(\mathbf{u}_{t+1})\big) + \big(f_T(\mathbf{u}_T) - f_T(\mathbf{u}_{T+1})\big) \\
&= \sum_{t=1}^{T-1} \big(M(\mathbf{u}_{t+1}, \mathbf{v}_{t+1}) - M(\mathbf{u}_{t+1}, \mathbf{v}_t)\big) + \big(f_1(\mathbf{u}_1) - f_T(\mathbf{u}_{T+1})\big) \\
&\leq \sum_{t=1}^{T-1} L\|\mathbf{v}_{t+1} - \mathbf{v}_t\| + \big(f_1(\mathbf{u}_1) - f_T(\mathbf{u}_{T+1})\big) \\
&\leq LTd_2/\sqrt{2T} + \big(f_1(\mathbf{u}_1) - f_T(\mathbf{u}_{T+1})\big) \\
&\leq Ld_2\sqrt{T}/\sqrt{2} + 2C .
\end{aligned}
$$

where the fourth line uses $f_t(\cdot) := M(\cdot, \mathbf{v}_t)$, the fifth line uses the Lipschitz continuity of $M$. And the sixth line used the stability of the $\mathbf{v}_t$'s due to Lemma 2. Finally, we use $|f_t(u)| = |M(u, v_t)| \leq C$. □

### A.2 PROOF OF THEOREM 2

*Proof.* Writing explicitly $f_t(\mathbf{u}) := M(\mathbf{u}, \mathbf{v}_t)$ and $g_t(\mathbf{v}) := M(\mathbf{u}_t, \mathbf{v})$, and plugging these into the regret guarantees of $\mathcal{A}_1, \mathcal{A}_2$, we have,

$$
\sum_{t=1}^{T} M(\mathbf{u}_t, \mathbf{v}_t) - \min_{\mathbf{u}\in\mathcal{K}_1} \sum_{t=1}^{T} M(\mathbf{u}, \mathbf{v}_t) \leq B_T^{\mathcal{A}_1} , \tag{5}
$$

$$
\sum_{t=1}^{T} -M(\mathbf{u}_t, \mathbf{v}_t) - \min_{\mathbf{v}\in\mathcal{K}_2} \sum_{t=1}^{T} -M(\mathbf{u}_t, \mathbf{v}) \leq B_T^{\mathcal{A}_2} . \tag{6}
$$

By definition, $\min_{\mathbf{u}\in\mathcal{K}_1} \sum_{t=1}^{T} M(\mathbf{u}, \mathbf{v}_t) \leq \mathbb{E}_{\mathbf{u}\sim\mathcal{D}_1}[\sum_{t=1}^{T} M(\mathbf{u}, \mathbf{v}_t)]$. Using this together with Equation (5), we get,

$$
\sum_{t=1}^{T} M(\mathbf{u}_t, \mathbf{v}_t) - \mathbb{E}_{\mathbf{u}\sim\mathcal{D}_1}[\sum_{t=1}^{T} M(\mathbf{u}, \mathbf{v}_t)] \leq B_T^{\mathcal{A}_1} , \tag{7}
$$

Summing Equations (6),(7), and dividing by $T$, we get,

$$
\max_{\mathbf{v}\in\mathcal{K}_2} \frac{1}{T} \sum_{t=1}^{T} M(\mathbf{u}_t, \mathbf{v}) - \mathbb{E}_{\mathbf{u}\sim\mathcal{D}_1}[\frac{1}{T} \sum_{t=1}^{T} M(\mathbf{u}, \mathbf{v}_t)] \leq \frac{B_T^{\mathcal{A}_1}}{T} + \frac{B_T^{\mathcal{A}_2}}{T} . \tag{8}
$$

Recalling that $\mathcal{D}_1 \sim \text{Uni}\{\mathbf{u}_1, \ldots, \mathbf{u}_t\}, \mathcal{D}_2 \sim \text{Uni}\{\mathbf{v}_1, \ldots, \mathbf{v}_t\}$, and denoting $\varepsilon := \frac{B_T^{\mathcal{A}_1}}{T} + \frac{B_T^{\mathcal{A}_2}}{T}$, we conclude that,

$$
\mathbb{E}_{(\mathbf{u},\mathbf{v})\sim\mathcal{D}_1\times\mathcal{D}_2}[M(\mathbf{u}, \mathbf{v})] \geq \max_{\mathbf{v}\in\mathcal{K}_2} \mathbb{E}_{\mathbf{u}\sim\mathcal{D}_1}[M(\mathbf{u}, \mathbf{v})] - \varepsilon .
$$

We can similarly show that,

$$
\mathbb{E}_{(\mathbf{u},\mathbf{v})\sim\mathcal{D}_1\times\mathcal{D}_2}[M(\mathbf{u}, \mathbf{v})] \leq \min_{\mathbf{u}\in\mathcal{K}_1} \mathbb{E}_{\mathbf{v}\sim\mathcal{D}_2}[M(\mathbf{u}, \mathbf{v})] + \varepsilon .
$$

which concludes the proof. □

# B  REMAINING PROOFS

## B.1  PROOF OF PROPOSITION 1

*Proof.* Look at the first term in the GAN objective, $\mathbb{E}_{p_{\text{data}}} \log h_{\mathbf{v}}(\mathbf{x})$. For a fixed $x$ we have,

$$\log h_{\mathbf{v}}(x) = -\log\left(1 + \exp(-\mathbf{v}^{\top} x)\right),$$

and it can be shown that the above expression is always concave in $\mathbf{v}$ [8]. Since an expectation over concave functions is also concave, this implies the concavity of the first term in $\mathcal{H}$.

Similarly, look at the second term in the GAN objective, $\mathbb{E}_{\mathbf{z} \in p_{\mathbf{z}}} \log(1 - h_{\mathbf{v}}(G_{\mathbf{u}}(\mathbf{z})))$. For a fixed $G_{\mathbf{u}}(z)$ we have,

$$\log(1 - h_{\mathbf{v}}(G_{\mathbf{u}}(\mathbf{z}))) = -\log\left(1 + \exp(+v^{\top} G_{\mathbf{u}}(\mathbf{z}))\right)$$

and it can be shown that the above expression is always concave in $\mathbf{v}$. Since an expectation over concave functions is also concave, this implies the concavity of the second term in $\mathcal{H}$.

Thus $\mathcal{H}$ is a sum of two concave terms and is therefore concave in $\mathbf{v}$. □

## B.2  PROOF OF LEMMA 1

*Proof.* Writing explicitly $f_t(\mathbf{u}) := M(\mathbf{u}, \mathbf{v}_t)$ and $g_t(\mathbf{v}) := M(\mathbf{u}_t, \mathbf{v})$, and plugging these into the regret guarantees of $\mathcal{A}_1, \mathcal{A}_2$, we have,

$$\sum_{t=1}^{T} M(\mathbf{u}_t, \mathbf{v}_t) - \min_{\mathbf{u} \in \mathcal{K}_1} \sum_{t=1}^{T} M(\mathbf{u}, \mathbf{v}_t) \leq B_T^{\mathcal{A}_1},$$

$$\sum_{t=1}^{T} -M(\mathbf{u}_t, \mathbf{v}_t) - \min_{\mathbf{v} \in \mathcal{K}_2} \sum_{t=1}^{T} -M(\mathbf{u}_t, \mathbf{v}) \leq B_T^{\mathcal{A}_2}.$$

Summing the above equations and dividing by $T$, we get,

$$\max_{\mathbf{v} \in \mathcal{K}_2} \frac{1}{T} \sum_{t=1}^{T} M(\mathbf{u}_t, \mathbf{v}) - \min_{\mathbf{u} \in \mathcal{K}_1} \frac{1}{T} \sum_{t=1}^{T} M(\mathbf{u}, \mathbf{v}_t) \leq \frac{B_T^{\mathcal{A}_1}}{T} + \frac{B_T^{\mathcal{A}_2}}{T} := \varepsilon. \tag{9}$$

Next we show that the second term above is always smaller than the minimax value,

$$\min_{\mathbf{u} \in \mathcal{K}_1} \frac{1}{T} \sum_{t=1}^{T} M(\mathbf{u}, \mathbf{v}_t) \leq \min_{\mathbf{u} \in \mathcal{K}_1} \frac{1}{T} \sum_{t=1}^{T} \max_{\mathbf{v} \in \mathcal{K}_2} M(\mathbf{u}, \mathbf{v})$$

$$= \min_{\mathbf{u} \in \mathcal{K}_1} \max_{\mathbf{v} \in \mathcal{K}_2} M(\mathbf{u}, \mathbf{v})$$

Plugging the above into Equation (9), and recalling $\mathcal{D}_1 \sim \text{Uni}\{\mathbf{u}_1, \ldots, \mathbf{u}_t\}$, we get,

$$\max_{\mathbf{v} \in \mathcal{K}_2} \mathbb{E}_{\mathbf{u} \sim \mathcal{D}_1} M(\mathbf{u}, \mathbf{v}) \leq \min_{\mathbf{u} \in \mathcal{K}_1} \max_{\mathbf{v} \in \mathcal{K}_2} M(\mathbf{u}, \mathbf{v}) + \varepsilon.$$

which concludes the proof. □

## B.3  PROOF OF THE SECOND PART OF LEMMA 2 (STABILITY OF FTRL SEQUENCE IN CONCAVE CASE)

*Proof.* Here we establish the stability of the FTRL decision rule, $\mathcal{A}_2$, depicted in Theorem 1.

Note that the following applies to this FTRL objective,

$$\sum_{\tau=0}^{t-1} \nabla g_\tau(\mathbf{v}_\tau)^{\top} \mathbf{v} - \frac{\sqrt{T}}{2\eta_0} \|\mathbf{v}\|^2 = -\frac{\sqrt{T}}{2\eta_0} \left\| \mathbf{v} - \frac{\eta_0}{\sqrt{T}} \sum_{\tau=0}^{t-1} \nabla g_\tau(\mathbf{v}_\tau) \right\|^2 + C \tag{10}$$

---

[8]For $a \in \mathbb{R}$, the 1-dimensional function $Q(a) = -\log\left(1 + \exp(-a)\right)$ is concave. Note that $\log h_{\mathbf{v}}(x)$ is a composition of $Q$ over a linear function in $\mathbf{v}$, and is therefore concave.

Where $C$ is a constant independent of $\mathbf{v}$.

Let us denote by $\Pi_{\mathcal{K}_2}$ the projection operator onto $\mathcal{K}_2 \subset \mathbb{R}^n$, meaning,

$$\Pi_{\mathcal{K}_2}(\mathbf{v}_0) = \min_{\mathbf{v} \in \mathcal{K}_2} \|\mathbf{v}_0 - \mathbf{v}\|, \qquad \forall \mathbf{v}_0 \in \mathbb{R}^n$$

By Equation (10) the FTRL rule, $\mathcal{A}_2$, can be written as follows,

$$\mathbf{v}_t = \operatorname*{argmin}_{\mathbf{v} \in \mathcal{K}_2} \left\| \mathbf{v} - \frac{\eta_0}{\sqrt{T}} \sum_{\tau=0}^{t-1} \nabla g_\tau(\mathbf{v}_\tau) \right\|^2$$

$$= \Pi_{\mathcal{K}_2} \left( -\frac{\eta_0}{\sqrt{T}} \sum_{\tau=0}^{t-1} \nabla g_\tau(\mathbf{v}_\tau) \right).$$

The projection operator is a contraction (see e.g, Hazan et al. (2016)), using this together with the above implies,

$$\|\mathbf{v}_{t+1} - \mathbf{v}_t\| \leq \left\| \Pi_{\mathcal{K}_2} \left( -\frac{\eta_0}{\sqrt{T}} \sum_{\tau=0}^{t} \nabla g_\tau(\mathbf{v}_\tau) \right) - \Pi_{\mathcal{K}_2} \left( -\frac{\eta_0}{\sqrt{T}} \sum_{\tau=0}^{t-1} \nabla g_\tau(\mathbf{v}_\tau) \right) \right\|$$

$$\leq \left\| -\frac{\eta_0}{\sqrt{T}} \nabla g_t(\mathbf{v}_t) \right\|$$

$$\leq d_2/\sqrt{2T}.$$

where we used $\|\nabla g_t(\mathbf{v}_t)\| \leq L$ which is due to the Lipschitz continuity of $M$. We also used $\eta_0 = d_2/\sqrt{2}L$. $\qquad \square$

## C  PRACTICAL CHEKHOV GAN ALGORITHM

The pseudo-code of algorithms $\mathcal{A}_1$ and $\mathcal{A}_2$ is given in Algorithm 3. The algorithm is symmetric for both players and consists as follows. At every step $t$ if we are currently in the switching mode (i.e. $t \mod m == 0$) and the queue is full, we remove a model from the end of the queue, which is the oldest one. Otherwise, we do not remove any model from the queue, but instead just override the head (first) element with the current update.

---

**Algorithm 3** Update queue for Algorithm $\mathcal{A}_1$ and $\mathcal{A}_2$

---
    **Input**: Current step $t$, $m > 0$
    **if** ($t \mod m == 0$ and $|\mathcal{Q}|) == K$) **then**
        $\mathcal{Q}$.remove_last()
        $\mathcal{Q}$.insert($f_t$)
        $m = m + inc$
    **else**
        $\mathcal{Q}$.replace_first($f_t$)
    **end if**

---

We set the initial spacing, $m$, to $\frac{N}{K}$, where $N$ is the number of update steps per epoch, and $K$ is the number of past states we keep. The number of updates per epoch is just the number of the data points divided by the size of the minibatches we use. The default value of $inc$ is 10. Depending on the dataset and number of total update steps, for higher values of $K$, this is the only parameter that needs to be tuned. We find that our model is not sensitive to the regularization hyperparameters. For symmetric architectures of the generator and the discriminator (such as DCGAN), for practitioners, we recommend using the same regularization for both players. For our experiments we set the default regularization to 0.1.

## D  EXPERIMENTS

### D.1  TOY DATASET: MIXTURE OF GAUSSIANS

We perform several experiments on a toy dataset where we varied the architecture size and the sampling probabilities. The toy dataset consists of a mixture of 7 Gaussians with a standard deviation

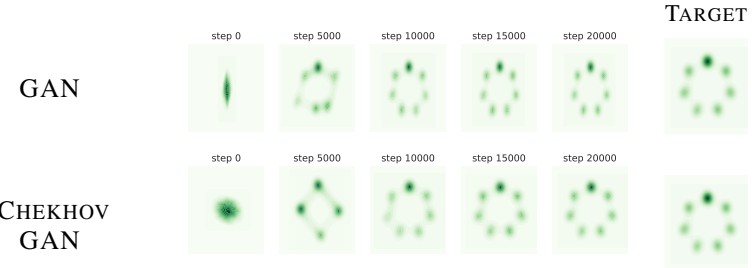

Figure 5: Both GAN and CHEKHOV GAN converge to the true data distribution when the dimensionality of the noise vector is 2

of 0.01 and means equally spaced around a unit circle.

The architecture for the generator consists in two fully connected layers (of size 128) and a linear projection to the dimensionality of the data (i.e. 2). The activation functions for the fully connected layers are tanh. The discriminator is symmetric and hence, composed of two fully connected layers (of size 128) followed by a linear layer of size 1. The activation functions for the fully connected layers are tanh, whereas the final layer uses sigmoid as an activation function.

Following Metz et al. (2016), we intialize the weights for both networks to be orthogonal with scaling of 0.8. AdamKingma & Ba (2014) was used as an optimizer for both the discriminator and the generator, with a learning rate of $1e-4$ and $\beta_1 = 0.5$. The discriminator and generator respectively minimize and maximize the objective

$$\mathbb{E}_{x \sim p_{\text{data}}(\mathbf{x})} - \log(D(x)) - \mathbb{E}_{z \sim \mathcal{N}(0, I_{256})} \log(1 - D(G(z))). \tag{11}$$

The setup is the same for both models. For CHEKHOV GAN we use $K = 5$ past states with L2 regularization on the network weights using an initial regularization parameter of 0.01.

**Effect of the latent dimension.** We find that for the case where $z \sim \mathcal{N}(0, I_{256})$, GANs with the traditional updates can fail to cover all modes by either rotating around the modes (as shown in Metz et al. (2016)) or converge to only a subset of the modes. However, if we sample the latent code from a lower dimensional space, e.g. $z \sim N(0, I_2)$, such that it matches the data dimensionality, the generator needs to learn a simpler mapping. We then observe that both GAN and CHEKHOV GAN are able to recover the true data distribution in this case (see Figure 5). Additionally, tuning the momentum of the optimiser can also stabilize the training of GANs, resulting in a generator that learns all the modes from the data distribution.

**Experiments with concave discriminator.** As out theoretical guarantees hold for a general (i.e. deep) generator and a concave discriminator, we perform experiments using a concave discriminator as well. We use the same architecture as described previously. In order to make the discriminator concave, we keep the parameters of the first two layers of the discriminator fixed and update only the parameters of the last layer. We generally find this much harder to stabilize due to the great imbalance between the number of trainable parameters of the discriminator and generator. Increasing the size of the last layer of the discriminator significantly improves the performance. Figure 6 shows that CHEKHOV GAN can learn the true data distribution. The final layer is increased by 10 times and the number of past states, $K$, is 10.

**Mode collapse.** We run an additional experiment directly targeted at testing for mode collapse. We sample points $x$ from the data distribution $p_{\text{data}}$ with different probabilities for each mode. We perform an experiment with 5 Gaussian mixtures, again of standard deviation 0.01 arranged in a circle. The probabilities to sample points from each of the modes are [0.35, 0.35, 0.1, 0.1, 0.1]. In this case two modes have higher probability and could potentially attract the gradients towards them

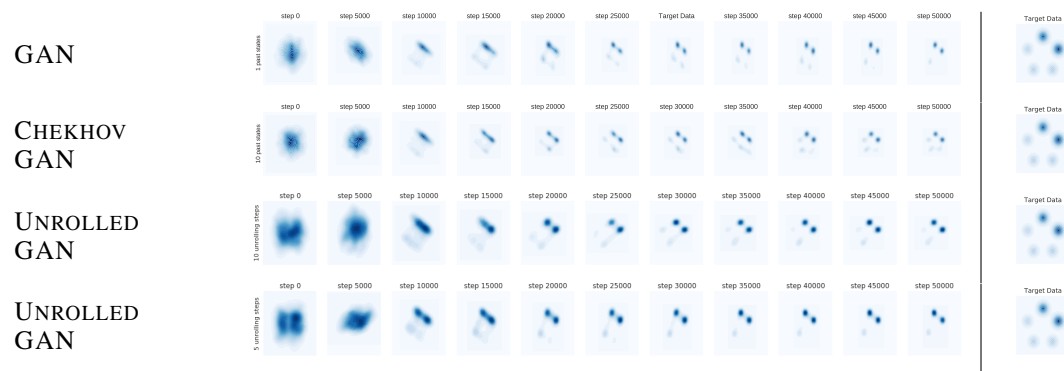

Figure 6: CHEKHOV GAN with concave discriminator is able to learn the toy data distribution with $K = 10$.

and cause mode collapse. In order to make things harder, the size of the hidden layers is 16 instead of 128. CHEKHOV GAN with $K = 10$ manages to recover the true data distribution in this case as well, unlike vanilla GANs. Using the history helps CHEKHOV GAN to spread its mass and cover even the modes with low sampling probability. Unrolled GAN Metz et al. (2016) instead uses the way the discriminator would react to the current changes when updating the generator. A comparison to Unrolled GANs and vanilla GANs is given in Figure 7.

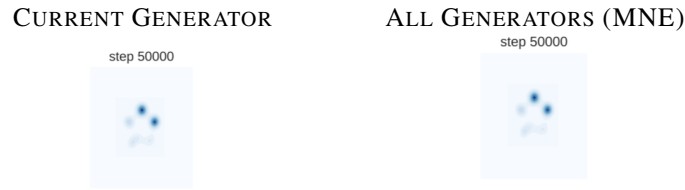

GAN

CHEKHOV GAN

UNROLLED GAN

UNROLLED GAN

Figure 7: Mode Collapse on a Gaussian Mixture. Comparison between a) vanilla GANs; b) CHEKHOV GAN with $K = 10$, increase=1 and regularization of 0.0005; c) Unrolled GAN with 10 unrolling steps and d) Unrolled GAN with 5 unrolling steps.

**Mixed Nash Equilibrium.** For all the experiments we generate from the most current generator only. We empirically find that generating such that we first sample uniformly at random one of the generators from the past, and then generating from it (which corresponds to the MNE) does not lead to any significant improvement (Figure 8). We hypothesize that this is due to a single generator being powerful enough to learn the distribution when playing against the experts from the past.

CURRENT GENERATOR          ALL GENERATORS (MNE)

step 50000                      step 50000

Figure 8: Comparison of CHEKHOV GAN for $K = 10$ when generating from the most current and all of the generators at the final step of training.

## D.2 AUGMENTED MNIST

We here detail the experiment on the Stacked MNIST dataset. The dataset is created by stacking three randomly selected MNIST images in the color channels, resulting in a 3-channel image that belongs to one out of 1000 possible classes. The architectures of the generator and discriminator are given in Table 6 and Table 7, respectively.

Table 6: Stacked MNIST: Generator Architecture

| Layer | Number of outputs |
|---|---|
| Input $z \sim N(0, I_{256})$ | |
| Fully Connected | 512 (reshape to [-1, 4, 4, 64] ) |
| Deconvolution | 32 |
| Deconvolution | 16 |
| Deconvolution | 8 |
| Deconvolution | 3 |

Table 7: Stacked MNIST: Discriminator Architecture

| Layer | Number of outputs |
|---|---|
| Convolution | 4 |
| Convolution | 8 |
| Convolution | 16 |
| Flatten and Fully Connected | 1 |

We use a simplified version of the DCGAN architecture as suggested by Metz et al. (2016). It contains "deconvolutional layers" which are implemented as transposed convolutions. All convolutions and deconvolutions use kernel size of $3 \times 3$ with a stride of 2. The weights are initialized using the Xavier initialization Glorot & Bengio (2010). The activation units for the discriminator are leaky ReLUs with a leak of 0.3, whereas the generator uses standard ReLUs. We train all models for 20 epochs with a batch size of 32, using the RMSProp optimizer with batch normalization. The optimal learning rate for GAN is 0.001, and for CHEKHOV GAN is 0.01. For all CHEKHOV GAN models we use regularization of 0.1 for the discriminator and 0.0001 for the generator. The regularization is L2 regularization only on the fully connected layers. For $K = 5$, the increase parameter *inc* is set to 50. For K=10, *inc* is 120.

## D.3   CIFAR10 / CELEBA

We use the full DCGAN architecture Radford et al. (2015) for the experiments on CIFAR10 and CelebA, detailed in Table 8 and Table 9.

Table 8: CIFAR10/CelebA Generator Architecture

| Layer | Number of outputs |
|---|---|
| Input $z \sim N(0, I_{256})$ | |
| Fully Connected | 32,768 (reshape to [-1, 4, 4, 512] ) |
| Deconvolution | 256 |
| Deconvolution | 128 |
| Deconvolution | 74 |
| Deconvolution | 3 |

Table 9: CIFAR10/CelebA Discriminator Architecture

| Layer | Number of outputs |
|---|---|
| Convolution | 64 |
| Convolution | 128 |
| Convolution | 256 |
| Convolution | 512 |
| Flatten and Fully Connected | 1 |

As for MNIST, we apply batch normalization. The activation functions for the generator are ReLUs, whereas the discriminator uses leaky ReLUs with a leak of 0.3. The learning rate for all the models is 0.0002 for both the generator and the discriminator and the updates are performed using the Adam optimizer. The regularization for CHEKHOV GAN is 0.1 and the increase parameter *inc* is 10.

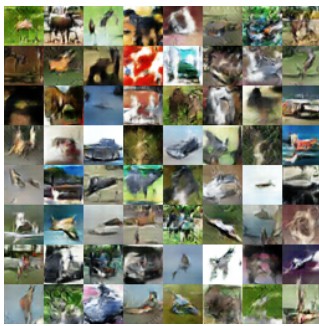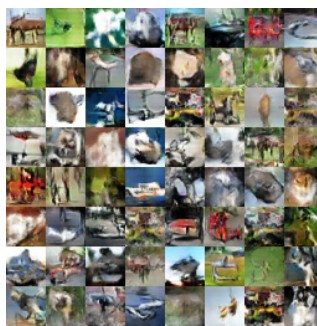

Figure 9: Random batch of generated images from GAN after training for 30 epochs (on the left) and CHEKHOV GAN (K=25) after training for 30 epochs (on the right)

### D.3.1 RESULTS ON CIFAR10

We train for 30 epochs, which we find to be the optimal number of training steps for vanilla GAN in terms of MSE on images from the validation set. Table 10 includes comparison to other baselines. The first set of baselines (given with purple color) consist of GAN where the updates in the inner loop (for the discriminator), the outer loop (for the generator), or both are performed 25 times. The baselines shown with green color are regularized versions of GANs, where we apply the same regularization as in our CHEKHOV GAN in order to show that the gain is not due to the regularization only. Figure 9 presents two randomly sampled batches from the generator trained with GAN and CHEKHOV GAN .

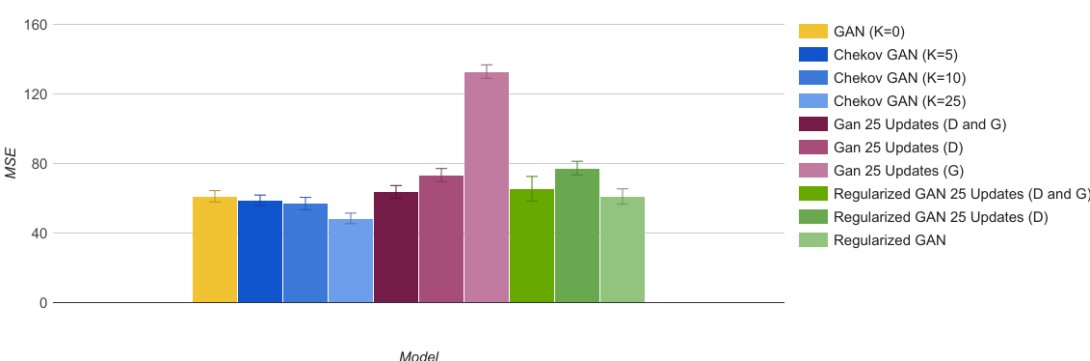

Table 10: CIFAR10: MSE for other baselines on target images that come from the training set. GAN 25 updates indicates that either the generator, the discriminator or both have been updated 25 times at each update step. Regularized GAN is vanilla GAN where the fully connected layers have regularization of 0.05.

Table 11 gives comparison between CHEKHOV GAN and Unrolled GAN for different numbers of past states or unrolling steps.

| Method | 5 states/steps | 10 states/steps | 25 states/steps |
|---|---|---|---|
| CHEKHOV GAN | $58.84 \pm 3.67$ | $56.99 \pm 3.49$ | $48.42 \pm 2.99$ |
| Unrolled GAN | $61.44 \pm 7.06$ | $55.60 \pm 5.52$ | - |

Table 11: CIFAR10: Comparison of average MSE on the training set between Unrolled GAN and CHEKHOV GAN for 5 and 10 unrolling steps and past states, respectively. The numbers for Unrolled GAN are taken from Metz et al. (2016).

### D.3.2   CelebA

All models are trained for 10 epochs. Randomly generated batches of images are shown in Figure 10.

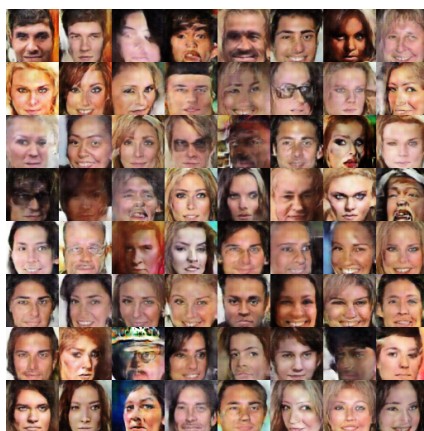 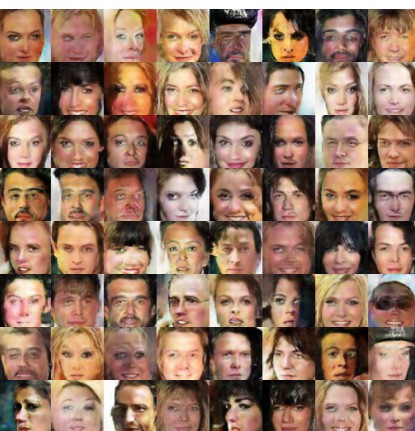

Figure 10: Random batch of generated images from GAN (left) and CHEKHOV GAN ($K = 5$) (right) after training for 10 epochs.

### D.3.3   Details about Inference via Optimization on CIFAR10

This approach consists in finding a noise vector $z_{closest}$ that when used as input to the generator would produce an image that is the closest to a target image in terms of mean squared error (MSE):

$$z_{closest} = \underset{z}{\operatorname{argmin}} \operatorname{MSE}(G(z), x_{target}) \qquad\qquad x_{closest} = G(z_{closest}).$$

We report the MSE in image space between $x_{closest}$ and $x_{target}$. This measures the ability of the generator to generate samples that look like real images. A model engaging in mode collapse would fail to generate (approximate) images from the real data. Conversely, if a model covers the true data distribution it should be able to generate any specific image from it.

### D.4   Inception Score and FID on Cifar10

Additional plots that showcase inception scores for Chekhov WGAN and the WGAN-based baselines (WGAN, WGAN GP, MIX+WGAN) and Chekhov GAN and the GAN-based baselines (GAN, Unrolled GAN, MIX+GAN) are given in Figure 11.

For Chekhov WGAN we set the regularization to 0.3 and $inc = 1$, whereas for Chekhov GAN we set the regularization to 0.005 and $inc$ to 50. All the baselines were run using their recommended set of hyperparameters.

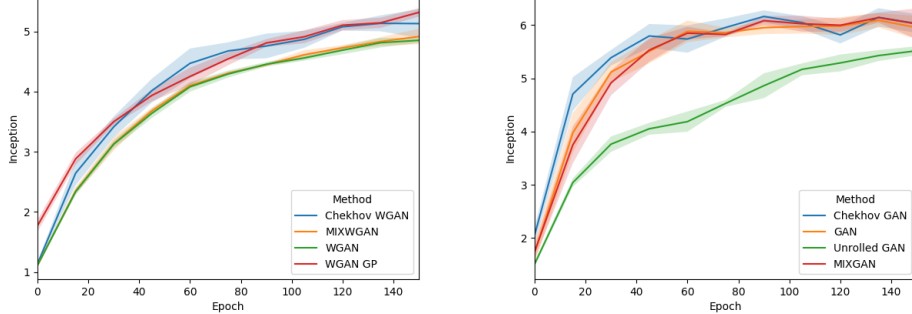

Figure 11: Comparison of Inception score: (left) WGAN-based variants, (right) GAN-based variants.The shaded area denotes the standard deviation.

