# OpenReview forum: "An Online Learning Approach to Generative Adversarial Networks"
_ICLR.cc/2018/Conference — Accept (Poster)_

### Official Review · AnonReviewer1 · 2017-11-28
**Powerful theory tools applied naturally to GAN dynamics**

**Rating:** 8
**Confidence:** 5

**Review:**

This is an interesting paper, exploring GAN dynamics using ideas from online learning, in particular the pioneering "sparring" follow-the-regularized leader analysis of Freund and Schapire (using what is listed here as Lemma 4). By restricting the discriminator to be a single layer, the maximum player plays over a concave (parameter) space which stabilizes the full sequence of losses so that Lemma 3 can be proved, allowing proof of the dynamics' convergence to a Nash equilibrium. The analysis suggests a practical (heuristic) algorithm incorporating two features which emerge from the theory: L2 regularization and keeping a history of past models. A very simple queue for the latter is shown to do quite competitively in practice.

This paper merits acceptance on theoretical merits alone, because the FTRL analysis for convex-concave games is a very robust tool from theory (see also the more recent sequel [Syrgkanis et al. 2016 "Fast convergence of regularized learning in games"]) that is natural to employ to gain insight on the much more brittle GAN case. The practical aspects are also interesting, because the incorporation of added randomness into the mixed generation strategy is an area where theoretical justifications do motivate practical performance gains; these ideas could clearly be developed in future work.

---

> ### Author Response · Authors · 2018-01-03
> **Thank You**
>
> Dear Reviewer,
>
> Thank you for your detailed and supportive review!

---

### Official Review · AnonReviewer2 · 2017-12-01
**borderline paper**

**Rating:** 7
**Confidence:** 4

**Review:**

It is well known that the original GAN (Goodfellow et al.) suffers from instability and mode collapsing. Indeed, existing work has pointed out that the standard GAN training process may not converge if we insist on obtaining pure strategies (for the minmax game). The present paper proposes to obtain mixed strategy through an online learning approach. Online learning (no regret) algorithms have been used in finding an equilibrium for zero sum game. However, most theoretical convergence results are known for convex-concave loss. One interesting theoretical contribution of the paper is to show that convergence result can be proved if one player is a shallow network (and concave in M).In particular, the concave player plays the FTRL algorithm with standard L2 regularization term. The regret of concave player can be bounded using existing result for FTRL. The regret for the other player is more interesting: it uses the fact the adversary's strategy doesn't change too drastically. Then a lemma by Kalai and Vempala can be used. The theory part of the paper is reasonable and quite well written.

Based on the theory developed, the paper presents a practical algorithm. Compared to the standard GAN training, the new algorithm returns mixed strategy and examine several previous models (instead of the latest) in each iteration. The paper claims that this may help to prevent model collapsing.

However, the experimental part is less satisfying. From figure 2, I don't see much advantage of Checkhov GAN. In other experiments, I don't see much improvement neither (CIFAR10 and CELEBA).The paper didn't really compare other popular GAN models, especially WGAN and its improved version, which is already quite popular by now and should be compared with.

Overall, I think it is a borderline paper.

-------------------------
I read the response and the new experimental results regarding WGAN.
The experimental results make more sense now.
It would be interesting to see whether the idea can be applied to more recent GAN models and still perform better.
I raised my score to 7.

---

> ### Author Response · Authors · 2018-01-03
> **Please consider updated experimental results clearly showing the benefits of our approach**
>
> Dear Reviewer,
>
> We thank you for your feedback and for your positive review regarding the theoretical part of the paper. In the following we answer your concerns and requests.
>
> Your review indicates that you believe the experimental part does not demonstrate the benefits of our approach. We respectfully disagree and we kindly ask you to look again at the experiments. Moreover, 1) as you requested, we have added new results further highlighting the benefits of our approach compared to the baselines, and 2) we improved the visibility of Figure 2 which we believe might have confused the reviewer in the original submission since some colors might not have been easily readable.
>
> To be specific:
> -In Figure 2 (toy example), you can see that after 50K steps our method captures all of the modes while the standard GAN is missing one mode.
> -In Table 1 (MNIST),  you can see that our method (10 states) generates 26% more classes compared to standard GAN training, and stabilizes the training which can be observed through the reduced variance. Please note that our approach also does better with respect to the reverse KL measure.
> -In Table 2 (CIFAR 10), you can see that our methods consistently outperform the standard GAN training with respect to the MSE measure. Concretely, the MSE of our method (25 states) is lower by 20% than the MSE of the standard training method. In terms of number of images from the test/training set with the lowest reconstruction, GAN achieves 0%, whereas our best variant achieves 82.43% and 80.33%, respectively.
> -In Table 4 (CelebA), you can see that our method outperforms the standard GAN training with respect to the number of modes that are classified by the auxiliary discriminator as not real.
> Concretely, our method improves by a factor of 50%.
> This means that using an auxiliary discriminator based on our method, only 1400 images of the true test set were recognized by the auxiliary discriminator as fake.
> Conversely, using an auxiliary discriminator on the standard method, there were 3000 images recognized by the auxiliary discriminator as fake.
> - In addition, we have now added a new section with experiments showcasing Inception Score and Frechet Dirichlet Distance (FID) for our method and show improvement over several strong baselines (Section 5.3).
>
> “From figure 2, I don't see much advantage of Checkhov GAN”
> See our answer above, light colors indicate a mode with less mass. We therefore clearly observe that the normal GAN suffers from mode collapse, while our approach finds all the modes. We hope that by going again over the experimental part, you will find that our method does improve over the standard GAN training.
>
> Regarding comparison to other methods: WGAN is a different loss function rather than a different GAN training method. Our method can be applied to other GAN loss functions, including WGAN. Thus, WGAN is not a competitor to our approach, but rather another setup where our approach could be applied. We do agree that a comparison to WGAN is valuable and we have now added the requested results to the paper (please see Table 5, Figure 3, Figure 4 and Figure 11).
> *Applying our method on top of WGAN (denoted Chekhov WGAN in the paper), outperforms WGAN across all epochs consistently in terms of both metrics (inception score and FID).

---

### Official Review · AnonReviewer4 · 2017-12-13
**Good theory, but large gap with the practical proposal is not sufficiently bridged by experimental evidence**

**Rating:** 5
**Confidence:** 4

**Review:**

The paper applies tools from online learning to GANs. In the case of a shallow discriminator, the authors proved some results on the convergence of their proposed algorithm (an adaptation of FTRL) in GAN games, by leveraging the fact that when D update is small, the problem setup meets the ideal conditions for no-regret algorithms. The paper then takes the intuition from the semi-shallow case and propose a heuristic training procedure for deep GAN game.

Overall the paper is very well written. The theory is significant to the GAN literature, probably less so to the online learning community. In practice, with deep D, trained by single gradient update steps for G and D, instead of the "argmin" in Algo 1., the assumptions of the theory break. This is OK as long as sufficient experiment results verify that the intuitions suggested by the theory still qualitatively hold true. However, this is where I have issues with the work:

1) In all quantitative results, Chekhov GAN do not significantly beat unrolled GAN. Unrolled GAN looks at historical D's through unrolled optimization, but not the history of G. So this lack of significant difference in results raise the question of whether any improvement of Chekhov GAN is coming from the online learning perspective for D and G, or simply due to the fact that it considers historical D models (which could be motivated by sth other than the online learning theory).

2) The mixture GAN approach suggested in Arora et al. (2017) is very related to this work, as acknowledged in Sec. 2.1, but no in-depth analysis is carried out. I suggest the authors to either discuss why Chekhov GAN is obviously superior and hence no experiments are needed, or compare them experimentally.

3) In the current state, it is hard to place the quantitative results in context with other common methods in the recent literature such as WGAN with gradient penalty. I suggest the authors to either report some results in terms of inception scores on cifar10 with similar architectures used in other methods for comparison. Alternatively please show WGAN-GP and/or other method results in at least one or two experiments using the evaluation methods in the paper.

In summary, almost all the experiments in the paper are trying to establish improvement over basic GAN, which would be OK if the gap between theory and practice is small. But in this case, it is not. So it is not entirely convincing that the practical Algo 2 works better for the reason suggested by the theory, nor it drastically improves practical results that it could become the standard technique in the literature.

---

> ### Author Response · Authors · 2018-01-03
> **Updated version includes the requested experimental results**
>
> Dear Reviewer,
>
> Thank you for your valuable feedback and the positive review regarding the theoretical part of the paper. We have added a significant number of experimental results to address your concerns and suggestions about the experimental part of the paper. We detail the changes made to the paper below.
>
> 1) Unrolled GAN: We first would like to point out that, unlike what you mentioned in your review, unrolled GAN does not look at historical D’s but rather it looks at **future** discriminator’s updates by unrolling a few optimization steps, thus making the two algorithms very different. The unrolling procedure comes with significant drawbacks as one needs to compute future steps which will not be used to update the model parameters. In contrast, our approach makes use of past steps and is therefore less wasteful in terms of computation (note that this could be further sped-up using parallel computations, which cannot be done with Unrolled GAN). Experiments on toy datasets show the benefits of using the history (Chekhov GAN) in comparison to Unrolled GAN (Figure 7). We have now added more experimental results comparing to Unrolled GAN in terms of Inception Score and FID, where we can clearly see that our approach achieves significantly better scores (Table 5, Figure 4 and Figure 11).
>
> 2) The mixture GAN approach suggested in Arora et al: We agree with the reviewer that a comparison to this work is valuable and we have now added the requested results to the updated version of the paper (Table 5). These results demonstrate that our approach achieves better results than Arora et al. when both methods are applied on top of WGAN, despite having 5 times less trainable parameters. Applying both methods on top of GAN yields comparable scores, but with notably less variance for Chekhov GAN.
>
>
> 3) WGAN & WGAN-GP: As requested by the reviewer, we have added a comparison to WGAN and WGAN GP in Table 5 in the paper where we clearly see better scores for Chekhov GAN. Note that although we originally developed our approach for the vanilla GAN, a similar algorithm can be applied to the Wasserstein objective which is also a min-max objective. As a proof of concept, we also provide empirical results for our approach with the Wasserstein objective and show that it also consistently improves upon the baseline.
>
> In summary, Chekhov GAN outperforms GAN over various metrics (MSE, number of missing modes, reverse KL divergence, number of generated classes, missing modes due to catastrophic forgetting) and achieves comparable performance in terms of Inception Score and FID, with reduced variance. At the same time, we successfully apply our algorithm on top of WGAN and achieve consistent improvement. We further show improvement across several other baselines as well.
>
> As you have mentioned, our algorithm has theoretical guarantees which we believe are of significant interest for the GAN literature. The practical algorithm is strongly inspired by the theory (using the history of generators and discriminators and updating the parameters by taking a gradient step guided by the FTRL objective) and outperforms the baselines across several metrics and datasets. Closing the gap between the theory and practice would be an interesting direction for future work as there are clear theoretical benefits.

---

### Public Comment · ~Leon_Boellmann1 · 2017-12-05
**Gap between theory and practice**

 Dear authors,
 The paper proposes to train GANs using FTRL. However, in FTRL each step is a best-response action (see Eq. (3)). In practical training, the networks is trained using gradient descent . My question is that why does it not train the discriminator and the generator using multiple steps until convergence, which more corresponds to the theory?

Thanks!

---

> ### Author Response · Authors · 2018-01-03
> **More practical while retaining performance**
>
> Dear Leon,
>
> Thank you for your comment!
>
> There are two reasons for that:
> -First, we wanted to make a fair comparison to the standard training method, which uses one update rather than several upades per round.
> -Second, we have noticed that for our method, using several updates vs. a single update did not make a big difference. We therefore decided to present the experiments with a single update which is more practical.

---

### Public Comment · ~Leon_Boellmann1 · 2017-12-05
**Cannot avoid mode collapse in a toy example**

Dear authors,
 I have one question on how the proposed approach is able to avoid mode collapse.
 Consider this simple example: the data samples are all greater than 10, the initial generated samples are all all less than -10, the initial D(x) = 0 for x<=0 and D(x) = 1 for all x>0. By gradient descent, the generator will get stuck from the very beginning, i.e., the generator will never be updated. Then even if you use the follow-the-regularizer approach, the generator still cannot be updated. How do you resolve this issue?

 Thanks!

---

> ### Author Response · Authors · 2018-01-03
> **FTRL avoids mode collapse in the toy example**
>
> Dear Leon,
>
> Thank you for your interest!
>
> In the example that you have raised the (theoretic) FTRL approach will not lead to mode collapse.
> In order to see why this is the case, let’s formalize your example:
> -Assume that the true data is uniformly distributed between [10,12].
> -Also assume that the generator can choose a single parameter \mu which induces a uniform distribution between [\mu-1, \mu+1].
> -Assume that the discriminator may choose two parameters W and b; this induces the following classification rule:  “+1” if Wx+b>0, and “0” otherwise.
>
> In your example, the initial parameters are, \mu = -11, and W=1, b=0.
> Now, at the second round, the generator will choose a new \mu which minimizes its loss. In our case this would lead to some \mu>=1 (note that FTRL does not use gradient step, but rather full minimization).
> In the next round, the discriminator will change W and b in order to separate between the generated and true data (if possible). In the following round, the generator will again update its parameters and so on… This process will persist until \mu converges to \mu=11, which is the true data.
>
> The reason that the FTRL approach converges is that it does not rely only on gradient information but rather on more global information.
>
> Note that the example you raised is indeed a hurdle for the standard GAN optimization method, and all other approaches that rely on gradients, including our practical algorithm (unless we inject additive noise to the samples that we feed into the discriminator, which is a common practice for training GANs).
>
> The question you raised highlights the potential benefits of implementing the full FTRL approach, which we hope would inspire/be addressed in future work.

---

> > ### Public Comment · ~Leon_Boellmann1 · 2018-01-03
> > **Thanks!**
> >
> >  Dear author,
> >  Thanks a lot for your explanation. I understand that FTRL approach works in theory. Actually my toy example was meant to point out that the practical Chekhov GAN may not work because it is using gradients, instead of best-response action. Overall, I think your paper is very good, especially that many experiments have been done to demonstrate the performance. I would say the application of FTRL to GAN training itself is already a very good idea and is worth publishing. We came up a similar idea as yours, so that is probably why I think highly of your work. We actually prove that our scheme can converge to the optimal point without assuming semi-shallow architecture (under the same other conditions of your theorem).
> >
> > I would also point out that the gradient update and best-response action will result in completely different training dynamics. The best-response action would oscillate and averaging over the past is needed to make sure it converges. Actually the gradient update can guarantee the convergence of training as well without an extreme oscillation between the actions. Details can be found in "Gradient descent GAN optimization is locally stable". That is why I raised the next question of "gap between theory and practice".
> >
> > BTW, we also found a recent paper that applies gradient descents but manages to avoid mode collapse in the above-mentioned toy example as well. Actually I read many GAN papers that aim to resolve the mode collapse issue, but obviously does not work even for this simple example. I agree with the authors that the technique of injection of noise is possible to resolve this issue. In this sense, it is hard to distinguish whether the proposed method contributes to the good performance or the noise injection technique. Probably a more systematic approach to evaluate the GAN performance is needed. Thanks!

---

### Decision · Program_Chairs · 2018-01-29
**ICLR 2018 Conference Acceptance Decision**

**Decision:**

Accept (Poster)

**Comment:**

This paper presents a GAN training algorithm motivated by online learning. The method is shown to converge to a mixed Nash equilibrium in the case of a shallow discriminator. In the initial version of the paper, reviewers had concerns about weak baselines in the experiments, but the updated version includes comparisons against a variety of modern GAN architectures which have been claimed to fix mode dropping. This seems to address the main criticism of the reviewers. Overall, this paper seems like a worthwhile addition to the GAN literature.